# The 3D Printing of Biomass–Fungi Composites: Effects of Waiting Time after Mixture Preparation on Mechanical Properties, Rheological Properties, Minimum Extrusion Pressure, and Print Quality of the Prepared Mixture

Al Mazedur Rahman [1], Abhinav Bhardwaj [1], Zhijian Pei [1,*], Chukwuzubelu Ufodike [2] and Elena Castell-Perez [3]

1 Department of Industrial and Systems Engineering, Texas A&M University, College Station, TX 77843, USA
2 Department of Engineering Technology and Industrial Distribution, Texas A&M University, College Station, TX 77843, USA
3 Department of Biological and Agricultural Engineering, Texas A&M University, College Station, TX 77843, USA
* Correspondence: zjpei@tamu.edu

**Abstract:** Biomass–fungi composites, an emerging class of sustainable materials, have potential applications in the construction and packaging industries. Molding-based manufacturing methods are typically employed to make products from these composites. Recently, a 3D printing-based method was developed for biomass–fungi composites to eliminate the need for making molds and to facilitate customized product design compared with manufacturing methods based on molding and hot-pressing. This method has six stages: biomass–fungi material preparation; primary colonization; mixture preparation; printing; secondary colonization; and drying. This paper reports a study about the effects of waiting time between the mixture preparation and 3D printing using biomass–fungi composites. As the waiting time increased from 0.25 to 3 h, the hardness and compressibility of the prepared mixture increased. As the waiting time increased from 0.25 to 8 h, the shear viscosity showed a decreasing trend; the yield stress of the prepared mixture increased at the beginning, then significantly decreased until the waiting time reached 3 h, and then did not significantly vary after 3 h. As the waiting time increased, the storage modulus and loss modulus decreased, the loss tangent delta increased, and the minimum required printing pressure for continuous extrusion during extrusion-based 3D printing increased. The print quality (in terms of layer-height shrinkage and filament-width uniformity) was reasonably good when the waiting time did not exceed 4.5 h.

**Keywords:** biomass-fungi; 3D printing; mycelium; rheology; biocomposite; texture analysis

## 1. Introduction

Both the construction and packaging industries have a substantial negative impact on the environment [1]. The production of cement (widely used in the construction industry) and petroleum-derived plastics (widely used in the packaging industry) emits a huge amount of $CO_2$. In 2015, the U.S. construction sector generated 565.8 million metric tons of $CO_2$ emissions, making it the fourth highest $CO_2$-emitting sector in the U.S. [2]. The $CO_2$ emissions from fossil fuel extraction and transportation attributed to plastic production are about 9.5 to 10.5 million metric tons, respectively, per year in the U.S. [3]. The packaging industry consumes 38% of the petroleum-based plastics produced globally [4]. Based on the current trends in production and waste management, it is estimated that roughly 12 billion metric tons of plastic waste will be in the natural environment or landfills by 2050 [5]. Sustainable materials that do not emit $CO_2$ in their production and have biodegradable characteristics are desirable in the construction and packaging industries.

Biomass–fungi composites, as an alternative to petroleum-based plastics, are an emerging class of sustainable materials for the construction and packaging industries [6]. Lignocellulosic biomass (such as corn stalks, wheat straws, sugarcane leaves, and sorghum stalks) [7–18], used for producing biomass–fungi composites [19–21], is one of the most abundant sustainable raw materials on earth. In these composites, the biomass serves as a nutrition source for fungi; the fungi grow into a network of fine white filaments (mycelium) and bind the biomass together [22]. These composites generally have a low density, low cost, and tailorable biodegradability.

There are publications on biomass–fungi composites [22–27] and their applications [28–31]. These publications report the effects of fungal species, nutrient substrates, environmental growth conditions (e.g., light or dark and $CO_2$ concentration) on mycelium-based biocomposites, biocomposite fabrication methods [25,32,33], and their effects on the mechanical properties of fabricated samples [34]. Currently, molding-based methods are used to manufacture parts using biomass–fungi composites. Another method is based on hot-pressing. This method includes three steps: (1) fungal or enzymatic incubation; (2) preparation for hot-pressing; and (3) hot-pressing [35]. These two methods can produce denser products, but consume more energy and offer less flexibility in designing complex shapes in small quantities. Both fabrication methods restrict the access to oxygen needed for the fungi to grow through the biomass. The 3D printing of biomass–fungi composites could facilitate the manufacturing of parts with complex shapes (such as sandwich and topology-optimized structures) in art, architecture, interior design, and construction [6,27,36–40] that cannot be easily produced using molding-based or hot-pressing based methods.

The effects of the mixture composition on the rheological properties of a mixture have already been reported [22]. Using biomass–fungi composites, a 3D printing-based manufacturing method consisting of six stages (Figure 1) has been reported [24]. The red boxes in Figure 1 represent the outcome of each stage.

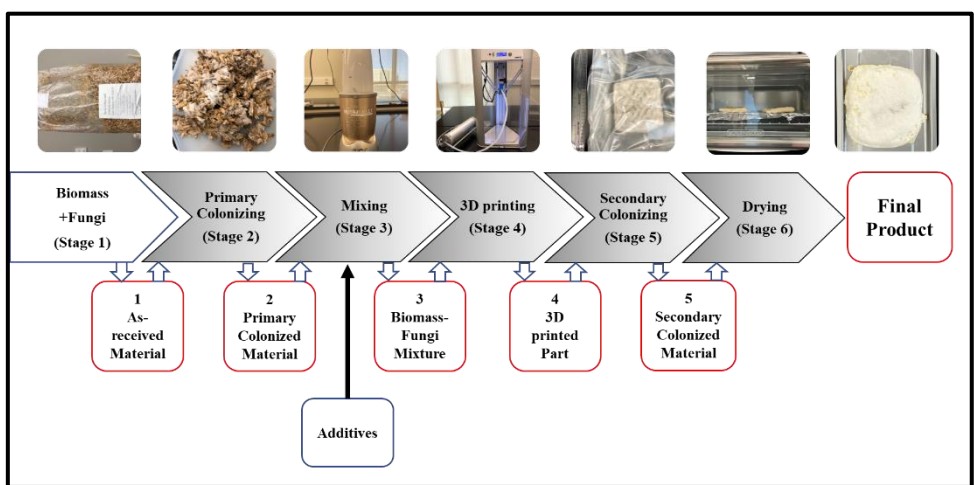

**Figure 1.** Six stages of the 3D printing-based method for biomass–fungi composites.

There are no reported studies regarding the effects of waiting time between mixing (mixture preparation for 3D printing) (Stage 3) and 3D printing with the prepared mixture (Stage 4) in the 3D printing of biomass–fungi composites. The purpose of this paper was to fill this gap in the literature. For extrusion-based 3D printing of biomass–fungi composites, it is necessary to know the effects of waiting time between mixing (Stage 3) and 3D printing (Stage 4) on the print quality. In this paper, we report a study about the effects of waiting time on the mechanical properties and rheological properties of a prepared mixture (at Stage 3), the minimum printing pressure required for continuous extrusion during extrusion-based 3D printing, and the quality of printed samples (in terms of layer-height shrinkage and filament-width uniformity). A texture profile analysis (TPA) was employed to measure the mechanical properties of the prepared mixture. Oscillatory

and steady shear experiments were conducted to evaluate the rheological properties of the prepared mixture, including viscosity, viscoelasticity, and yield stress [41–43]. These results were helpful in explaining the changes in print quality and minimum extrusion pressure of the prepared mixture.

## 2. Methodology

### 2.1. Materials

Biomass–fungi-mix material, wheat flour, water, and psyllium husk powder were used to prepare the biomass–fungi mixture. The biomass–fungi-mix material (Grow-it-Yourself: GROW.bio, Green Island, NY, USA) was received in a polypropylene bag with a filter (1.5 inch square in size) with 0.2 μm pores. The feedstock material underwent three steps in the factory. First, the biomass material was pasteurized to eliminate any harmful organisms. The pasteurized biomass material was then inoculated with fungi that belong to the Basidiomycete group and the fungi spread and multiplied in the biomass material. In the third step, this biomass–fungi-mix material was dehydrated before packaging [24].

Wheat flour (all-purpose flour: Great Value, Walmart, Bentonville, AR, USA) and psyllium husk (NOW Supplements, Bloomingdale, IL, USA) were procured from a local Walmart store.

The mixture was prepared batch by batch and each batch had 50 g of biomass–fungi-mix material. Every batch had the same composition as shown in Table 1. This composition was successfully used in a previous study [22].

**Table 1.** Mixture composition for each batch of 50 g biomass–fungi-mix material.

| Material | Amount |
| --- | --- |
| Biomass–fungi-mix material | 50 g $\pm$ 0.02 g |
| Wheat flour | 20 g $\pm$ 0.02 g |
| Water | 200 mL $\pm$ 0.02 mL |
| Psyllium husk powder | 10 g $\pm$ 0.02 g |

### 2.2. Experimental Procedures

The procedures (the six stages shown in Figure 1) described in a recent paper [22] were followed in this study. The biomass–fungi-mix material from Stage 1 underwent primary colonization in Stage 2. During Stage 3 (mixing), the primary colonized biomass–fungi material was mixed with wheat flour and water in a commercial mixer (NutriBullet PRO: Capital Brands, Los Angeles, CA, USA) for 15 s and turned into a liquid slurry. Psyllium husk powder was then added to the liquid slurry. The liquid slurry was manually mixed using a spoon for 1 to 2 min. The resultant biomass–fungi mixture was then ready for 3D printing (Stage 4).

Figure 2 shows the setup for the 3D printing experiments. A Delta 2040 (Delta 2040: WASP, Massa Lombarda RA, Italy) 3D printer was equipped with a storage container (volume: 3 L) to hold the mixture prepared in Stage 3. It had a pneumatically operated plastic piston that pushed the mixture through a custom-built nozzle assembly. An air compressor (Kobalt 4.3 gallon Electric Twin Stack Quiet Air Compressor: Kobalt, Mooresville, NC, USA) was used to generate the printing pressure on the piston in the storage container. A pressure gauge was used to measure the printing pressure. The nozzle assembly had a screw extruder and a custom-designed casing with a 6 mm × 6 mm nozzle opening. Stages 2 to 5 (primary colonizing, mixing, 3D printing, and secondary colonizing) were conducted at room temperature. All the apparatus was cleaned before and after every use with a 70% ethyl alcohol solution to avoid any contamination.

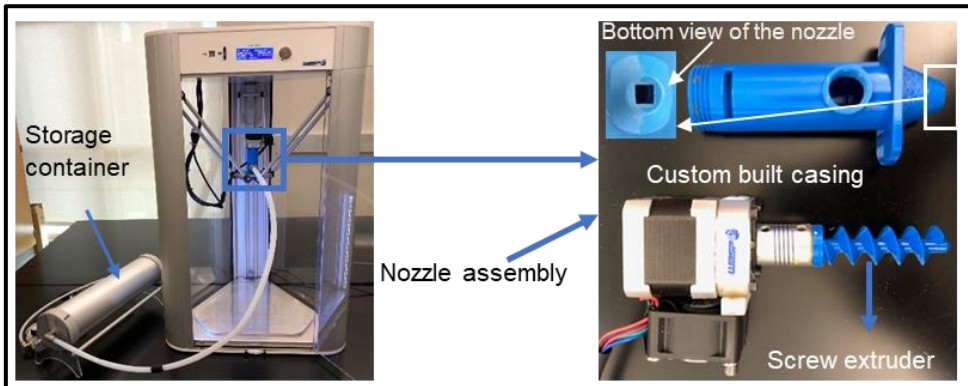

**Figure 2.** The 3D printing setup with custom-built nozzle assembly.

*2.3. Measurement of the Mechanical Properties of the Prepared Mixture*

A texture analyzer (TA.XT.Plus, Texture Technologies, Hamilton, MA, USA) was used to perform a texture profile analysis. This method has been extensively used to characterize biological materials, including food materials (such as fruit, whipped toppings, gels, and pudding desserts) [44–47]. Figure 3a shows the solid probe of the analyzer and a mixture sample. The cylindrical mixture samples were prepared using a cylindrical cup and a plunger, as shown in Figure 3b. The cup had an inner diameter of 15 mm and a height of 10 mm. The plunger had an outer diameter of 14.5 mm. The prepared biomass fungi mixture was filled in the cup and then the plunger was used to ensure there were no air pockets in the sample.

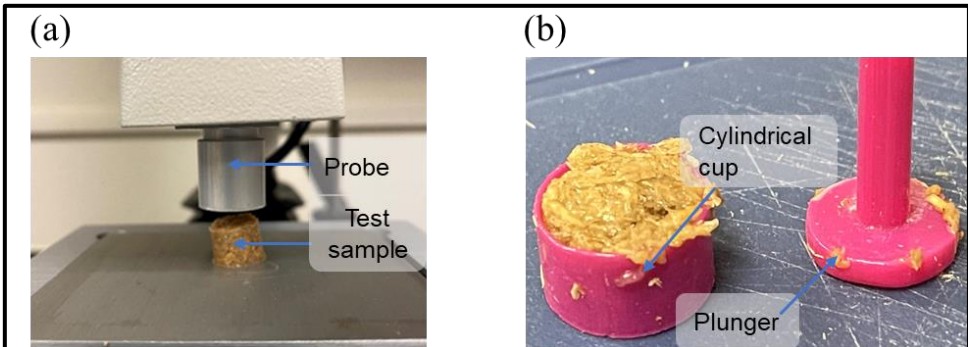

**Figure 3.** Texture analysis (**a**) measurement setup using TA.XT.Plus Texture Analyzer and (**b**) cup and plunger used to prepare mixture samples.

During the measurement, the sample was compressed by a solid probe from an initial height of 10 mm to 2 mm (80% strain) at a constant rate of 0.5 mm/s (during the pre-test, post-test, and test) and a trigger force of 5.0 g. The instrument began to record the data when the force reached the value of the trigger force. After the first compression cycle was completed, and after a 10 s delay, the sample was compressed again. After each test, a force-time (or force-distance) curve was obtained, as shown in Figure 4. From the resultant force-distance plot, the following mechanical properties could be obtained [48,49].

(1) Hardness (the maximum force recorded during the first compression cycle);
(2) Compressibility (the work required to deform the sample during the first compression and calculated as the area under the force-distance curve for the first compression cycle).

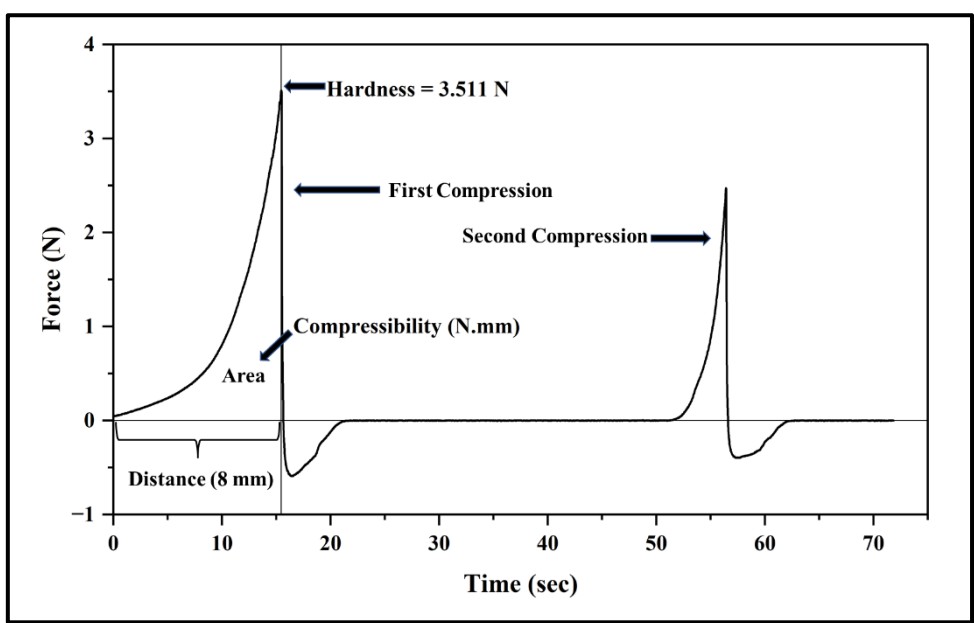

**Figure 4.** Force-time curve obtained from the texture analyzer for a waiting time of 1 min.

### 2.4. Measurement of the Rheological Properties of the Prepared Mixture

A rheometer (ARES G2, TA Instruments, New Castle, DE, USA) was used to measure the rheological properties of the prepared mixture. It had a pair of parallel plates 25 mm in diameter and a gap of 1 mm between the two plates (Figure 5). A forced convection oven was used to ensure that the temperature of the test environment remained constant at 25 °C. All the experiments were performed three times. To determine the shear viscosity of the prepared mixture, the shear rate was increased from 0.01 to 100 s$^{-1}$ and a linear viscoelastic range was observed with a strain sweep (0.1 to 100%) at a fixed angular frequency of 10 rad/s. A dynamic frequency sweep was performed at a constant strain of 1%. To obtain the yield stress (yield stress in rheology is defined as the applied stress at which irreversible plastic deformation is first observed across the sample), an oscillatory measurement was conducted at a constant frequency of 1 Hz and the stress was increased from 0.1 to 10$^4$ Pa.

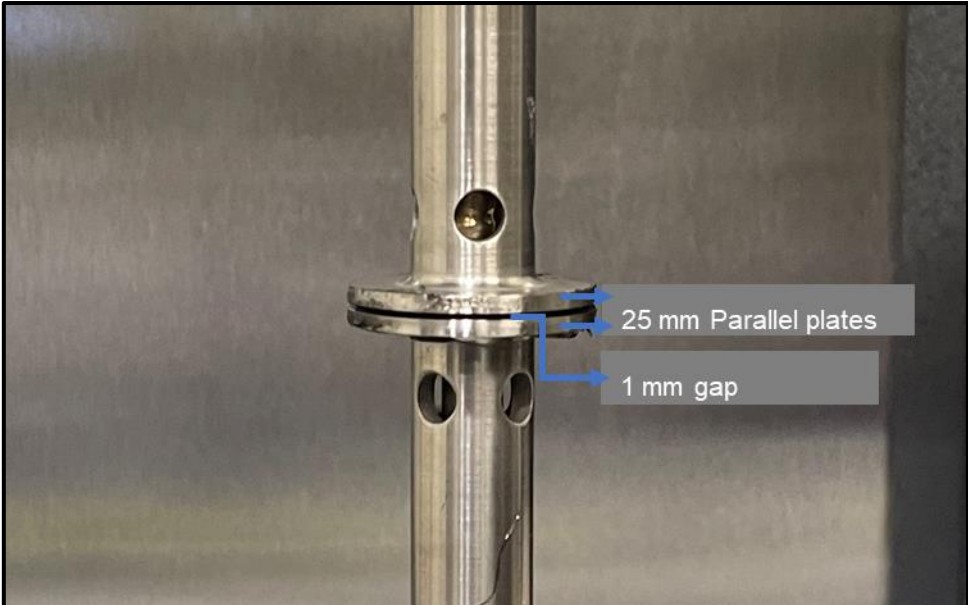

**Figure 5.** The pair of parallel plates of 25 mm diameter with a 1 mm gap in between.

### 2.5. Determination of the Minimum Printing Pressure Required for Continuous Extrusion

For all 3D printing experiments, the same design file was used to print the samples shown in Figure 5. The printing speed was 30 mm/s and layer height was 6 mm. The experiment at each level of waiting time was started at a printing pressure of 100 kPa. If continuous extrusion was not obtained (as shown in Figure 6a), the printing pressure was increased by an increment of 50 kPa. This continued until continuous extrusion (as shown in Figure 6b) was obtained. The printing pressure was recorded as the minimum printing pressure at this level of waiting time. If continuous extrusion was not obtained when the printing pressure was raised to 300 kPa, the printing pressure was increased by an increment of 20 or 40 kPa (instead of 50 kPa) for the subsequent printing.

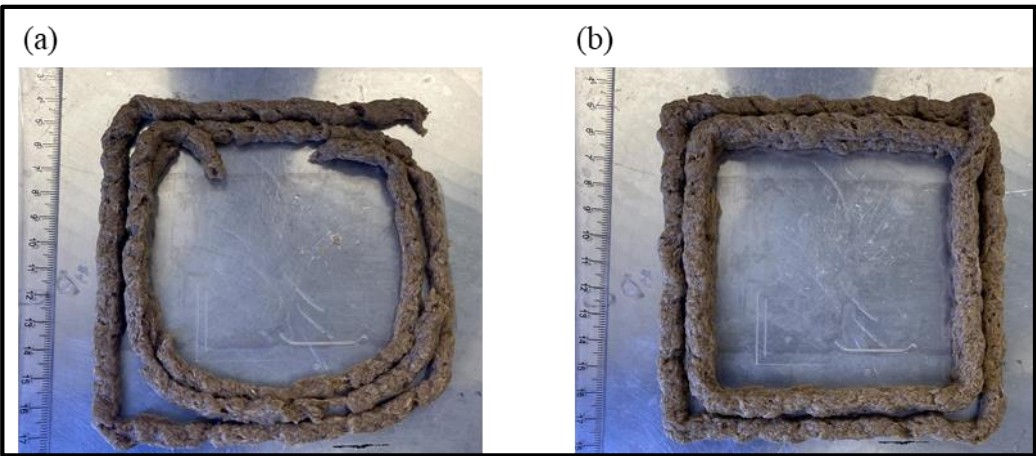

**Figure 6.** Printed samples by (**a**) discontinuous extrusion and (**b**) continuous extrusion.

### 2.6. Measurement of Print Quality

The print quality was evaluated using layer-height shrinkage and filament-width uniformity. A square-shaped sample, as shown in Figure 7a, was printed at each level of waiting time. The width and height of each of the four segments were measured using a slide caliper (Absolute Digimatic, Mitutoyo, Japan) at five different locations of each segment (as shown in Figure 7). The second layer of the square-shaped segments was then printed on top of the first layer, as shown in Figure 7b. The height of the four segments in the first layer was measured again. The layer-height shrinkage was calculated using the following equation:

$$\text{Height shrinkage} = \left( \frac{H_1 - H_2}{H_1} \right) * 100\% \tag{1}$$

where $H_1$ and $H_2$ are the average layer height of the four segments in the first layer before and after the second layer was printed, respectively. Each sample (Figure 7a) had four segments and five measurements were taken from each segment using a slide caliper (Absolute Digimatic, Mitutoyo, Japan). Filament-width uniformity was evaluated by a 95% confidence interval of the measured filament width data (a total of 20 data points at each level of waiting time).

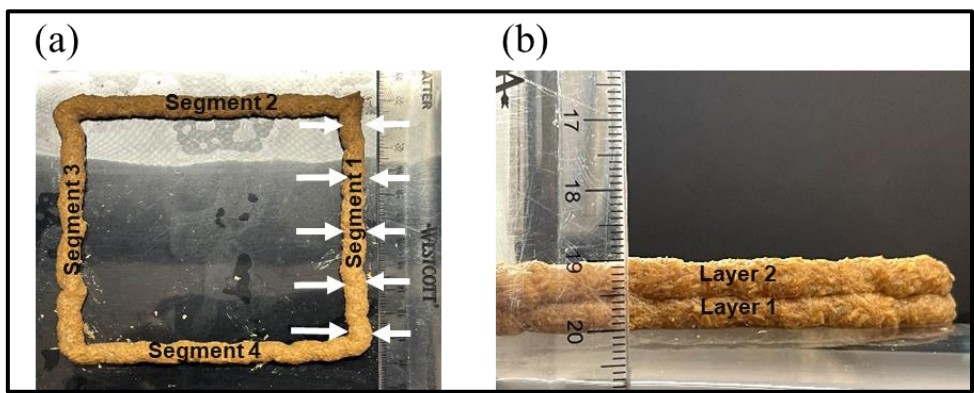

**Figure 7.** Printed samples for measurements of print quality: (**a**) one layer and (**b**) two layers.

*2.7. Statistical Analysis*

The differences in the variables (such as hardness and compressibility) caused by a change in the waiting time were examined by a t-test using Minitab software (Minitab, USA). The significance level ($\alpha$) used in these statistical analyses was 0.05. Therefore, when the *p*-value was smaller than 0.05, the difference was considered to be statistically significant.

## 3. Results and Discussion

*3.1. Mechanical Properties of the Prepared Mixture*

Table 2 shows the measured results on the hardness and compressibility of the prepared mixture at each level of waiting time. The presented values of the mean and 95% confidence interval at each level of waiting time were calculated from the measured data of at least four samples. Each experiment was performed with a different sample. As the waiting time increased from 0.25 to 3 h, the hardness significantly increased ($p < 0.05$) from 3.53 to 6 N and the compressibility significantly increased ($p < 0.05$) from 6.70 to 10.89 N.mm. The hardness values obtained for the prepared mixtures were comparable with the hardness values of other hydrogels reported in different studies. However, we noted that the other studies did not have the same probe diameter, compression speed, and deformation height [50–52]. Changes in the hardness and compressibility as the waiting time increased may have been caused by the swelling behavior of psyllium husk [53], water evaporation from the samples, and fungal growth inside the samples. A further investigation is needed to fully understand the causes.

**Table 2.** Hardness and compressibility of the prepared mixture at different levels of waiting time.

| Waiting Time (h) | Hardness (N) | Compressibility (N.mm) |
|---|---|---|
| 0.25 | 3.53 ± 0.58 | 6.70 ± 0.82 |
| 0.5 | 3.83 ± 0.41 | 6.90 ± 0.78 |
| 1 | 4.98 ± 1.05 | 9.05 ± 1.51 |
| 2 | 5.55 ± 0.32 | 9.64 ± 0.50 |
| 3 | 6.00 ± 0.81 | 10.89 ± 1.37 |
| 4 | 5.09 ± 0.63 | 9.15 ± 0.93 |
| 5 | 5.53 ± 0.34 | 10.24 ± 0.4 |
| 6.5 | 5.75 ± 0.86 | 10.25 ± 1.55 |
| 8 | 5.42 ± 0.63 | 9.82 ± 0.83 |

*3.2. Rheological Properties of the Prepared Mixture*

To study the effects of the waiting time on the rheological properties, nine levels of waiting time were selected: 0.25; 1; 2; 3; 4; 5; 6; 7; and 8 h. Figure 8 shows the viscosity vs. shear rate graphs for the nine levels of waiting time. In the figure, each data point (the mean and 5% standard deviation) was calculated from the measured data of three samples at each level of waiting time. It can be seen that the viscosity had a general decreasing

trend when the waiting time changed from 0.25 to 8 h. The prepared mixture should have exhibited shear-thinning behavior to enable a continuous flow through the nozzle [54,55]. This prepared biomass–fungi mixture showed pronounced shear-thinning behavior at all levels of waiting time. Its apparent viscosity decreased by several orders of magnitude as the shear rate increased from 0.01 to 100 s$^{-1}$. These shear rate values were close to the shear rate experienced by extruded material during extrusion-based 3D printing [56]. Similar shear-thinning properties were observed in another reported study on the printability of psyllium husk gel/gelatin blends [57].

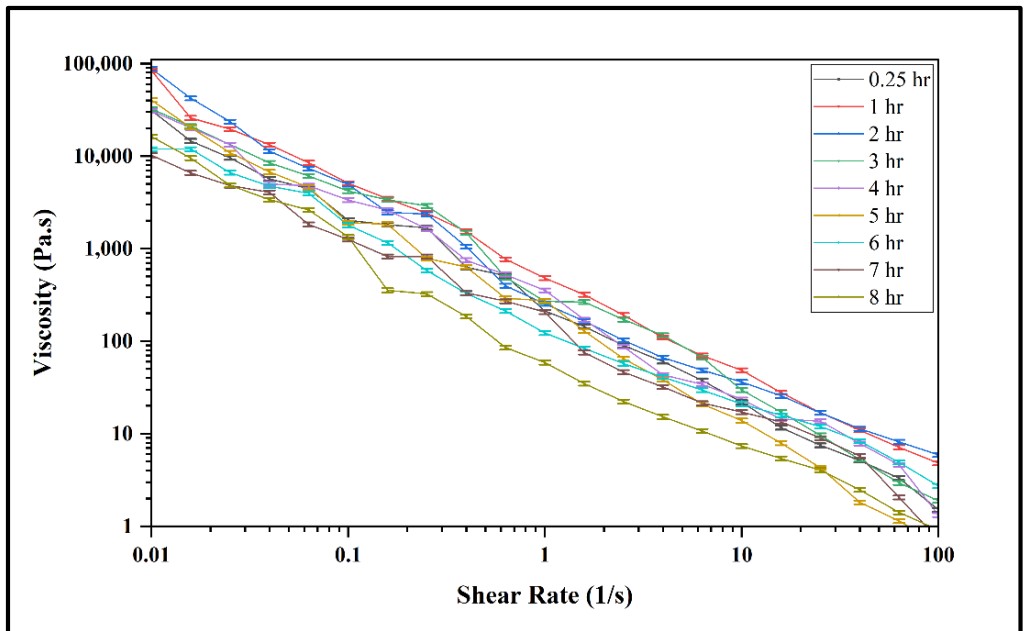

**Figure 8.** Viscosity vs. shear rate graphs for 9 levels of waiting time.

The yield stress of the prepared mixture should be high enough to maintain the overall structure after 3D printing and support the weight of the extruded material [58,59]. Figure 9 shows the yield stress data obtained from the stress sweep from 0.1 Pa to 10$^4$ Pa at different levels of waiting time. The yield stress significantly changed ($p < 0.05$) when the waiting time increased until it reached 3 h and did not considerably change ($p > 0.05$) as the waiting time increased from 3 to 8 h. The obtained yield stress values of the prepared mixture were comparable with the reported values of psyllium gels in the literature [60].

Figure 10 shows the storage modulus (Pa) vs. oscillation strain (%) graph obtained from the amplitude sweep. Viscoelastic properties such as the storage modulus, loss modulus, and loss tangent delta become strain-dependent at the end of the linear viscoelastic region. From the amplitude sweep, the end of the linear viscoelastic region was identified so that quality data could be obtained from the linear region. Initially, when the waiting time was 0.25 h, the linear viscoelastic region ended close to a 1% oscillation strain and then started to increase to around 10% as the waiting time increased to 8 h. A comparable behavior has been reported for psyllium husk gels [61].

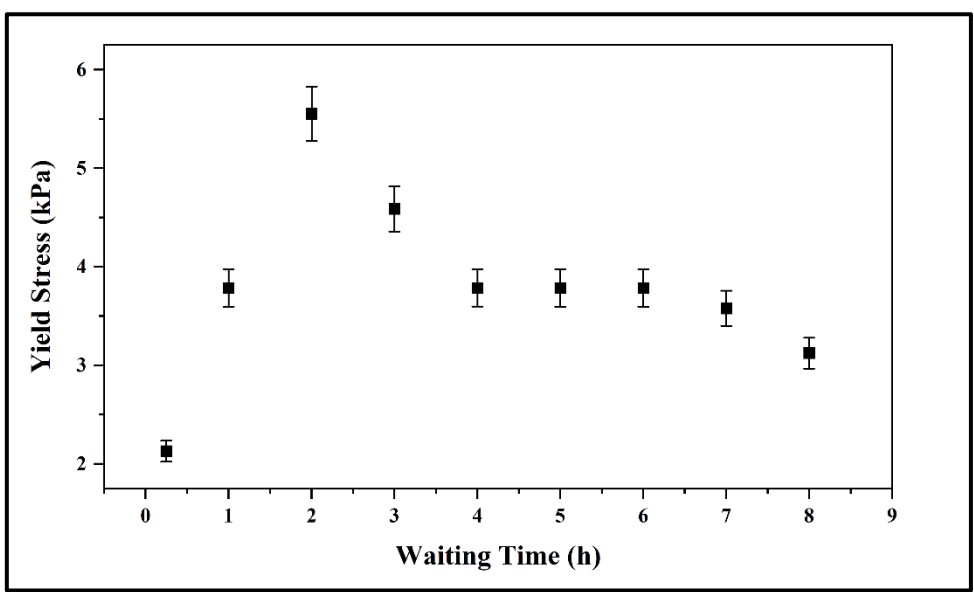

**Figure 9.** Yield stress data obtained from the stress sweep from 0.1 Pa to $10^4$ Pa at different levels of waiting time.

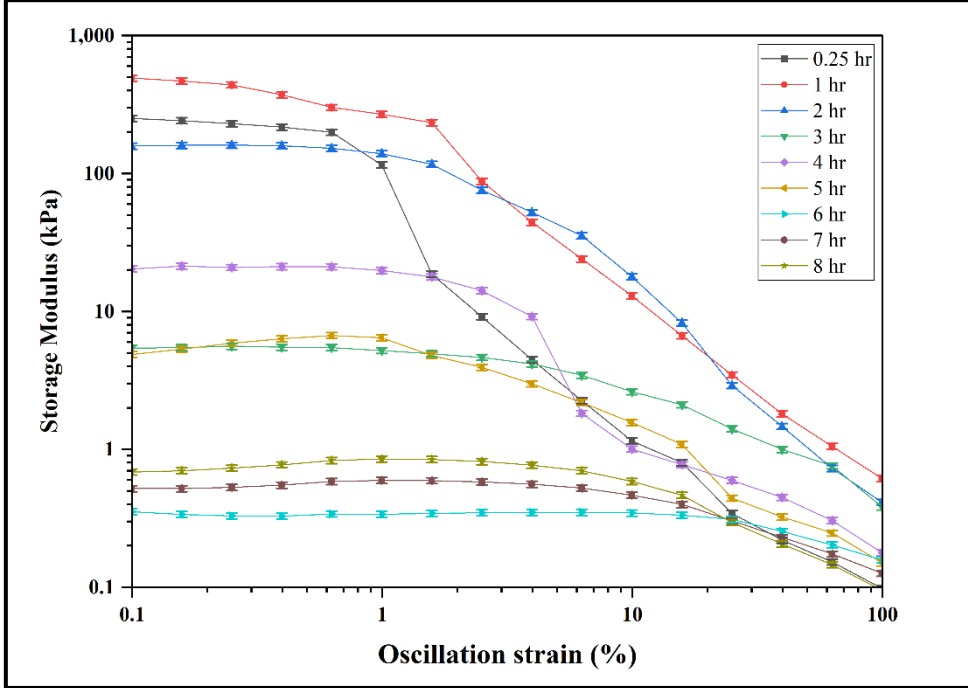

**Figure 10.** Storage modulus vs. oscillation strain graphs obtained from the amplitude sweep at 25 °C.

Figures 11–13 show the storage modulus (G′), loss modulus (G″), and loss tangent delta vs. angular frequency graphs, respectively. At all levels of waiting time, as the angular frequency increased, both the storage modulus and loss modulus increased, but the loss tangent delta decreased. As the waiting time increased, both the storage modulus and loss modulus decreased, but the loss tangent delta increased. Printing materials should be not only viscoelastic, but also elasticity-dominant (loss tangent delta < 1) to ensure the continuous extrusion of strands [62]. The prepared mixture should have rapidly recovered to a solid-like response with a sufficiently high storage modulus and yield stress to maintain its strand shape after 3D printing [54,55]. All the loss tangent delta values were less than 1 (Figure 13), which is required in order to ensure the continuous extrusion of strands in 3D printing [62]. If the prepared mixture had shown predominantly viscous behavior (G″ > G′),

it would have suggested that the mixture was not printable. The mixture at different levels of waiting time should have been printable, according to Figures 11 and 12. However, their ability to retain their filamentous shape deteriorated with an increase in waiting time as the storage modulus (Figure 11) significantly decreased ($p < 0.05$) [55]. When the waiting time reached 3 h, the changes in storage modulus and loss modulus became insignificant ($p > 0.05$); overlaps of a few graphs were observed, as shown in Figures 11 and 12.

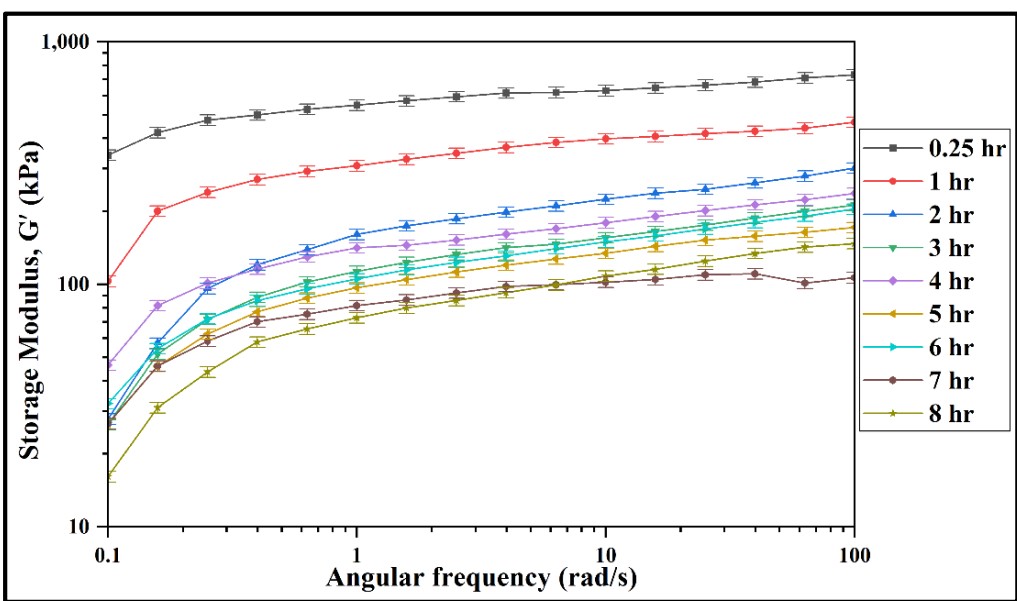

**Figure 11.** Storage modulus vs. angular frequency graphs at 9 different levels of waiting time.

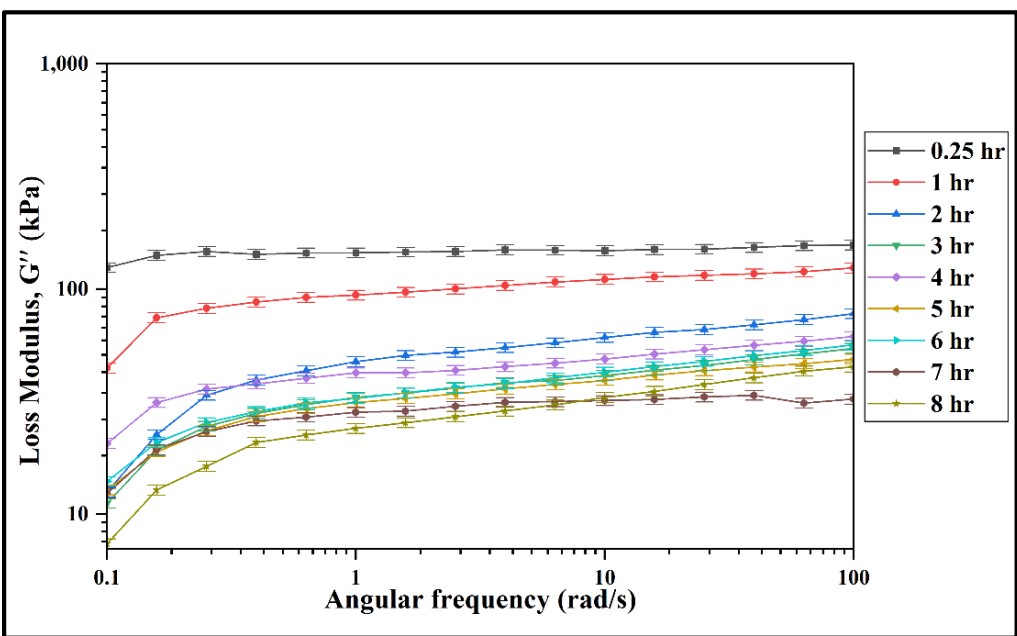

**Figure 12.** Loss modulus vs. angular frequency graphs at 9 different levels of waiting time.

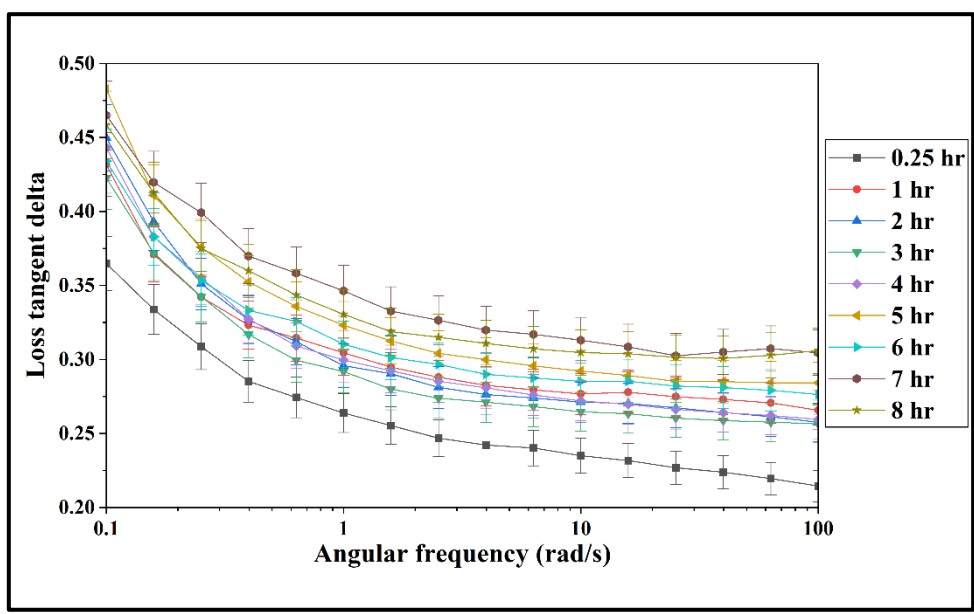

**Figure 13.** Loss tangent delta vs. angular frequency graphs at 9 different levels of waiting time.

### 3.3. Minimum Printing Pressure Required for Continuous Extrusion

Table 3 presents the experimental data of the minimum printing pressure required for continuous extrusion at each level of waiting time. As the waiting time increased to 3 h, the minimum printing pressure required for continuous extrusion increased rapidly to 360 kPa. When the waiting time was 3.5 h or longer, the minimum printing pressure remained approximately constant at 360 kPa. This phenomenon could be explained using texture analyzer data and rheological data. As the yield stress and hardness did not drastically change after the 3 h waiting time, a similar minimum required extrusion pressure was expected for the prepared mixture after 3.5 h of waiting time.

**Table 3.** Experimental data on the minimum required printing pressure at different levels of waiting time.

| Waiting Time (h) | Pressure (kPa) | | | | | | | |
|---|---|---|---|---|---|---|---|---|
| | 100 | 150 | 200 | 250 | 300 | 320 | 360 | 400 |
| 0.25 | 0 | 1 | 1 | 1 | 1 | 1 | 1 | 1 |
| 0.5 | 0 | 0 | 1 | 1 | 1 | 1 | 1 | 1 |
| 0.75 | 0 | 0 | 0 | 1 | 1 | 1 | 1 | 1 |
| 1 | 0 | 0 | 0 | 0 | 1 | 1 | 1 | 1 |
| 1.5 | 0 | 0 | 0 | 0 | 1 | 1 | 1 | 1 |
| 2 | 0 | 0 | 0 | 0 | 1 | 1 | 1 | 1 |
| 2.5 | 0 | 0 | 0 | 0 | 1 | 1 | 1 | 1 |
| 3 | 0 | 0 | 0 | 0 | 0 | 1 | 1 | 1 |
| 3.5 | 0 | 0 | 0 | 0 | 0 | 0 | 1 | 1 |
| 4 | 0 | 0 | 0 | 0 | 0 | 0 | 1 | 1 |
| 4.5 | 0 | 0 | 0 | 0 | 0 | 0 | 1 | 1 |
| 5 | 0 | 0 | 0 | 0 | 0 | 0 | 1 | 1 |
| 5.5 | 0 | 0 | 0 | 0 | 0 | 0 | 1 | 1 |
| 6.5 | 0 | 0 | 0 | 0 | 0 | 0 | 1 | 1 |
| 7.5 | 0 | 0 | 0 | 0 | 0 | 0 | 1 | 1 |
| 9 | 0 | 0 | 0 | 0 | 0 | 0 | 1 | 1 |

0 = Discontinuous, 1 = Continuous

### 3.4. Print Quality

3.4.1. Layer-Height Shrinkage

To study the effects of the waiting time on the print quality, six levels of waiting time were selected: 0.5; 1.5; 2.5; 4.5; 6.5; and 8.5 h. A minimum printing pressure was used at each level of waiting time. Figure 14 shows the change in layer-height shrinkage at each level of waiting time. The error bars represent the 95% confidence interval. It can be seen from the figure that the layer-height shrinkage increased as the waiting time increased from 0.5 to 4.5 h ($p < 0.05$); no significant changes were observed after 4.5 h of waiting time. The storage modulus data (Figure 11) showed that the storage modulus exhibited a decreasing trend as the waiting time increased. Changes in the storage modulus at different levels of waiting time might have been a cause for the changes in layer-height shrinkage when the waiting time increased.

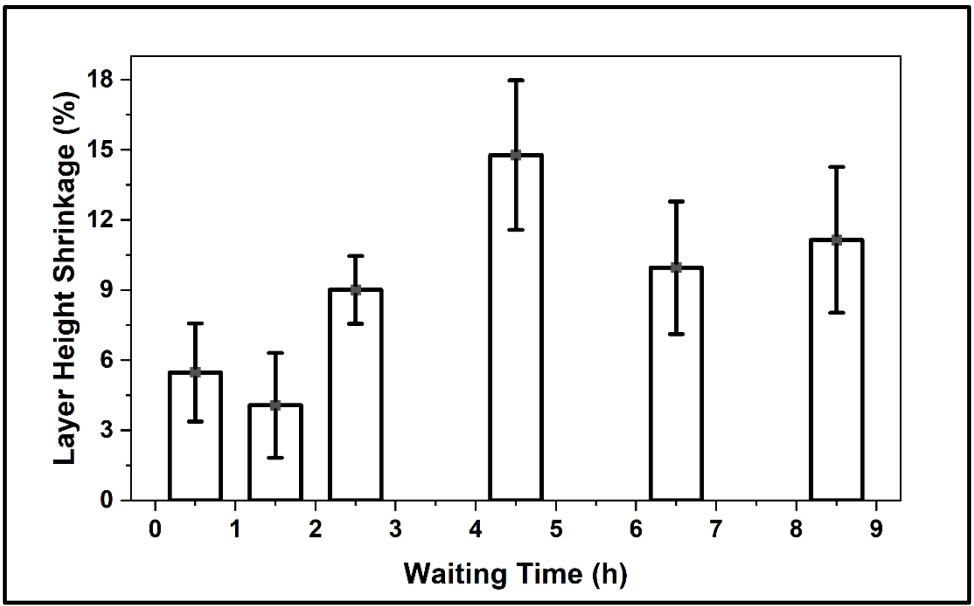

**Figure 14.** Effects of waiting time on layer-height shrinkage.

3.4.2. Filament-Width Uniformity

Figure 15 shows the effects of the waiting time on the filament-width uniformity (the 95% confidence interval of the measured data on the filament width). Figure 16 shows typical examples of the printed samples at different levels of waiting time.

As the waiting time increased from 0.5 to 2.5 h, the filament-width uniformity was relatively good (the 95% confidence interval of the measured data on the filament width was relatively small) and did not considerably change. As the waiting time increased to 4.5 h and longer, increased inconsistencies were observed (the 95% confidence interval of the measured data on the filament width became significantly larger). Figure 16f shows that the extruded filament became discontinuous.

The prepared mixture was used for 3D printing and secondary colonization at different levels of waiting time. During Stage 4 (secondary colonization), the printed samples were kept in filter patch bags and a box with an open lid for seven days. After three days, the samples were flipped so that an even fungal growth could be obtained throughout the whole printed material. The samples did not show any visible effect of different waiting times on the fungal growth (Figure 17). Future in-depth studies are required to confirm this initial observation.

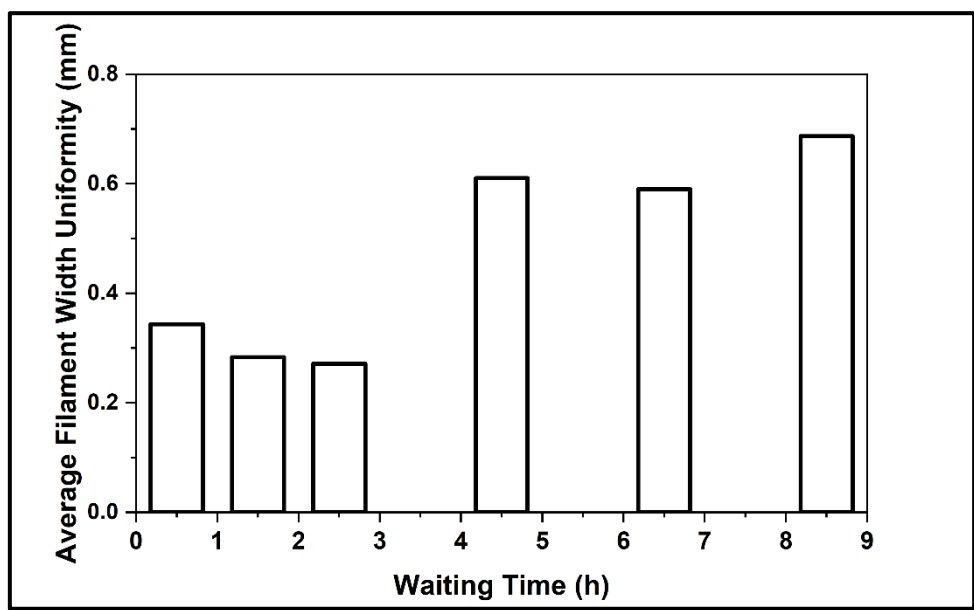

**Figure 15.** Effects of waiting time on filament width uniformity.

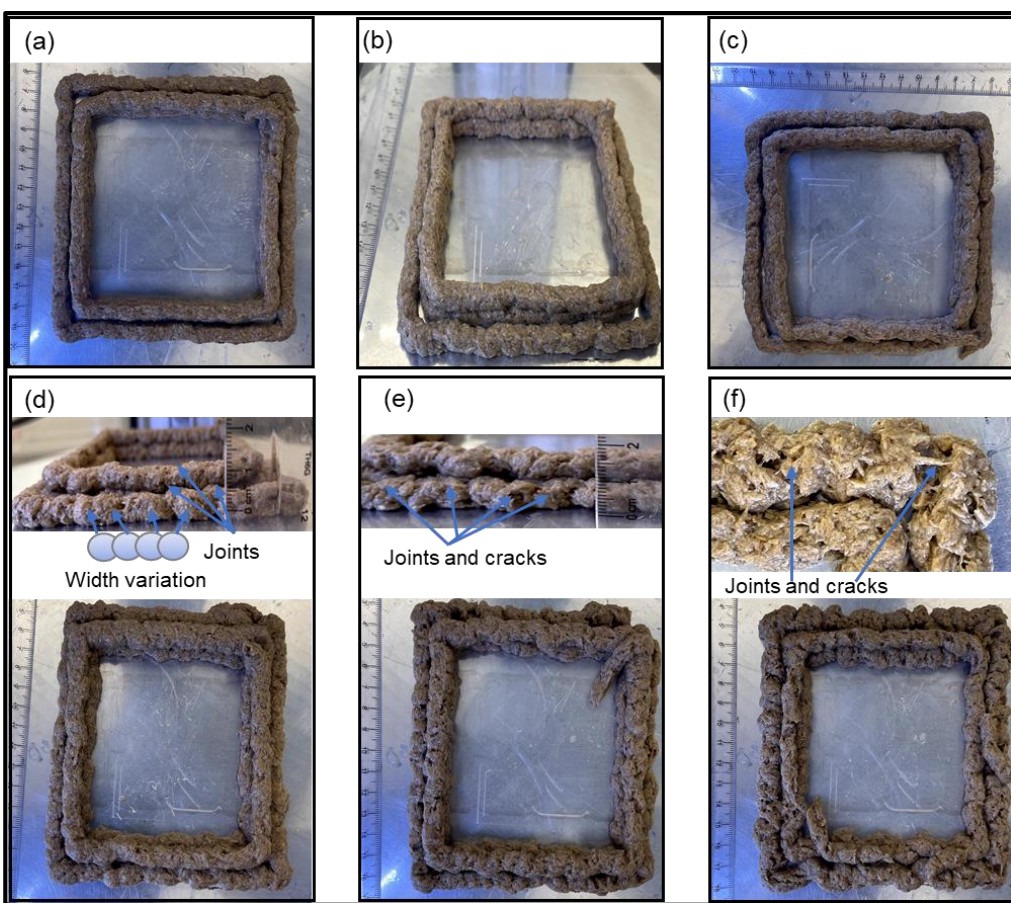

**Figure 16.** Samples printed at different levels of waiting time: (**a**) 0.5 h; (**b**) 1.5 h; (**c**) 2.5 h; (**d**) 4.5 h; (**e**) 6.5 h; (**f**) 8.5 h.

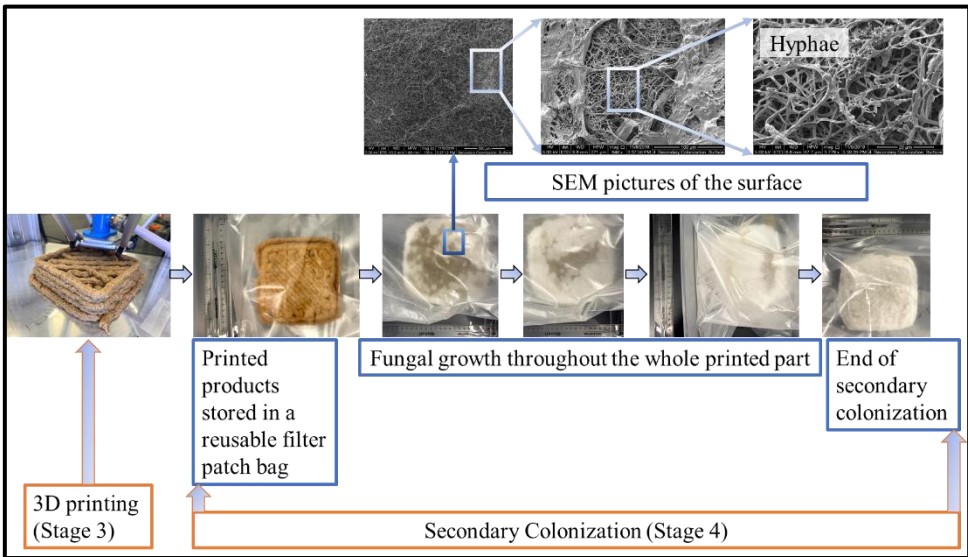

**Figure 17.** The 3D printing (Stage 3) and secondary colonization (Stage 4) using the prepared mixture from Stage 3.

### 4. Conclusions

In this paper, we report a study on the extrusion-based 3D printing of biomass–fungi composites. The input variable investigated was the waiting time between the mixture preparation and the 3D printing. The output variables were the hardness and compressibility of the prepared mixture as well as the rheological properties, minimum printing pressure required for continuous extrusion, and printing quality (in terms of layer-height shrinkage and filament-width uniformity). The main conclusions are stated below.

The hardness and compressibility of the prepared mixture significantly increased as the waiting time increased to 3 h, but did not significantly change as the waiting time increased further. The shear viscosity decreased as the waiting time increased from 0.25 to 8 h. The yield stress significantly changed as the waiting time increased to 3 h and did not considerably vary when the waiting time increased beyond 3 h. As the waiting time increased, both the storage modulus and the loss modulus decreased, but the loss tangent delta increased. The minimum printing pressure required for continuous extrusion increased as the waiting time increased to 3 h and remained at 360 kPa as the waiting time increased further. When the waiting time increased from 2.5 to 4.5 h, the layer-height shrinkage increased, and the filament-width uniformity deteriorated. Changes in the yield stress, storage modulus, loss modulus, hardness, and compressibility significantly affected the quality of the printed samples. The results showed that after preparing the mixture at Stage 3, the waiting time should not be longer than 4.5 h to obtain a good print quality. In the future, research could be conducted to understand how different additives, biomass substrates, and fungi species could change the effects of the waiting time. The rheological and mechanical properties of the prepared biomass–fungi mixture presented in this paper can be used as a reference for developing new printable biomass–fungi mixtures for 3D printing in the future.

**Author Contributions:** Methodology, A.M.R.; formal analysis, A.M.R. and A.B.; investigation, A.M.R., A.B. and E.C.-P.; writing—original draft preparation, A.M.R.; writing—review and editing, A.M.R., Z.P., C.U. and E.C.-P.; supervision, Z.P. All authors have read and agreed to the published version of the manuscript.

**Funding:** This research received no external funding.

**Data Availability Statement:** The authors confirm that the data to support the findings of this study are available within the article or upon request to the corresponding author.

**Conflicts of Interest:** The authors declare that they have no known competing financial interests or personal relationships that could have appeared to influence the work reported in this paper.

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
