# Peer review of "The 3D Printing of Biomass–Fungi Composites: Effects of Waiting Time after Mixture Preparation on Mechanical Properties, Rheological Properties, Minimum Extrusion Pressure, and Print Quality of the Prepared Mixture"

_jcs, doi:10.3390/jcs6080237_

Round 1
Reviewer 1 Report
The paper "3D Printing of Biomass-Fungi Composites: Effects of
Waiting Time after Mixture Preparation on Mechanical
Properties, Rheological Properties, Minimum Extrusion
Pressure and Print Quality of the Prepared Mixture" has been improved. In this form is suitable to be publish before minor revision.
The most important point is the interpretation of the viscosity obtained by oscillatory test. The values obtained with these kind of test are the complex viscosity vs. frequency that they can be considered equal to shear viscosity vs. shear rate only when Cox Merz rule is valid. I think that is not this the case because the material is clearly a suspension.
In addition i ask to the authors to add the strain sweep plot to show the Linear Viscosity Region. This information is very useful for the reader because the material is not standard.
Finally i think that the title can be improved by avoiding making only a list of tests
Author Response
The paper "3D Printing of Biomass-Fungi Composites: Effects of Waiting Time after Mixture Preparation on Mechanical Properties, Rheological Properties, Minimum Extrusion Pressure and Print Quality of the Prepared Mixture" has been improved. In this form is suitable to be publish before minor revision.
The most important point is the interpretation of the viscosity obtained by oscillatory test. The values obtained with these kind of test are the complex viscosity vs. frequency that they can be considered equal to shear viscosity vs. shear rate only when Cox Merz rule is valid. I think that is not this the case because the material is clearly a suspension.
The authors agree with the reviewer about using the Cox Merz rule. That’s why the authors only reported viscosity as a function of the shear rate changing from 0.01 to 100 s-1. The complex viscosity vs. frequency obtained from the oscillatory analysis is not reported in the paper.
In addition i ask to the authors to add the strain sweep plot to show the Linear Viscosity Region. This information is very useful for the reader because the material is not standard.
Authors do not understand whether the reviewer meant linear viscoelastic region or viscosity region. Figure 10 reports the storage modulus vs. strain sweep (0.1 – 100 %) at the fixed angular frequency of 10 rad/s to show the linear viscoelastic region. The authors think that the results shown in Section 3.2 (Rheological properties of the prepared mixture) provide a basic understanding of rheological properties of the biomass-fungi composite used in this paper.
Finally, i think that the title can be improved by avoiding making only a list of tests
The authors do not understand how the reviewer would like to improve the title. The current title is “3D Printing of Biomass-Fungi Composites: Effects of Waiting Time after Mixture Preparation on Mechanical Properties, Rheological Properties, Minimum Extrusion Pressure and Print Quality of the Prepared Mixture.” An alternative title can be “3D Printing of Biomass-Fungi Composites: Effects of Waiting Time on Print Quality of the Prepared Mixture.” The authors are fine with either of them. The editor can decide which one to use.

Reviewer 2 Report
Dear Authors,
Generally, this is a very interesting article about biocomposites production. Especially that the authors take into account the possibility of printing with the use of extruders. Since extrusion is a fast and continuous process, it is a good approach. However, I think you should justify the purpose of your research a little better. In fact, it is enough to add (one or two) supplementary sentences.
Moreover:
Figure 4 and table 2. Consider reporting the results in MPa. Theoretically, the values given in Newtons [N] are correct because you gave the dimensions of the sample cup. But were the samples always identical? Therefore, in my opinion, the value should be given in MPa. Consider it. The rest of the article in my opinion is well written.
Author Response
Generally, this is a very interesting article about biocomposites production. Especially that the authors take into account the possibility of printing with the use of extruders. Since extrusion is a fast and continuous process, it is a good approach. However, I think you should justify the purpose of your research a little better. In fact, it is enough to add (one or two) supplementary sentences.
The following sentences have been modified in Section 1 (Introduction) to clearly justify the purpose of the research:
There are no reported studies regarding the effects of waiting time between mixing (mixture preparation for 3D printing) (Stage 3) and 3D printing with the prepared mixture (Stage 4) in 3D printing based manufacturing method using biomass-fungi composites. The purpose of this paper is to fill this gap in the literature.
Figure 4 and table 2. Consider reporting the results in MPa. Theoretically, the values given in Newtons [N] are correct because you gave the dimensions of the sample cup. But were the samples always identical? Therefore, in my opinion, the value should be given in MPa. Consider it. The rest of the article in my opinion is well written.
The authors agree with the reviewer that the unit of hardness (for solid materials such as metals and ceramics) is usually P, MPa, or GPa (not N). However, for the texture analyzer used to measure the properties of soft materials (such as biological materials), the hardness is defined as following: “the maximum force does hereby present the hardness of the hydrogel formulation” ; “force required for a pre-determined deformation” [3-8]. In all the reported studies the authors have read, either gram (g) or Newton (N) was used as the unit of hardness for texture analysis.
References:
[1] Hurler, J., Engesland, A., Poorahmary Kermany, B., and Škalko‐Basnet, N., Improved texture analysis for hydrogel characterization: Gel cohesiveness, adhesiveness, and hardness. Journal of Applied Polymer Science, 2012. 125(1): p. 180-188.
[2] Kaya, S., Effect of salt on hardness and whiteness of Gaziantep cheese during short-term brining. Journal of Food Engineering, 2002. 52(2): p. 155-159.
[3] Ahmed, N., El Soda, M., Hassan, A., and Frank, J., Improving the textural properties of an acid-coagulated (Karish) cheese using exopolysaccharide producing cultures. LWT-Food science and technology, 2005. 38(8): p. 843-847.
[4] Jones, D.S., Woolfson, A.D., and Djokic, J., Texture profile analysis of bioadhesive polymeric semisolids: mechanical characterization and investigation of interactions between formulation components. Journal of applied polymer science, 1996. 61(12): p. 2229-2234.
[5] Jones, M., Mautner, A., Luenco, S., Bismarck, A., and John, S., Engineered mycelium composite construction materials from fungal biorefineries: A critical review. Materials & Design, 2020. 187: p. 108397.
[6] Nishinari, K., Fang, Y., and Rosenthal, A., Human oral processing and texture profile analysis parameters: Bridging the gap between the sensory evaluation and the instrumental measurements. Journal of Texture Studies, 2019. 50(5): p. 369-380.
[7] Pons, M. and Fiszman, S., Instrumental texture profile analysis with particular reference to gelled systems. Journal of texture studies, 1996. 27(6): p. 597-624.
[8] Trinh, K.T. and Glasgow, S. On the texture profile analysis test. in Proceedings of the Chemeca. 2012.

Reviewer 3 Report
I still think that this is an incremental work with respect to what is reported in the literature on the same topic and the novelty is therefore very limited.
"Optical microscopy": it means studying the surface morphology of the 3D-printed sample. This is not very difficult to do and could be of interest to the readers, as one of the main issues related to 3D printing is the quality of the surface of the obtained printed parts. Indeed, the morphology and surface quality of the 3D printed samples is completely disregarded,
Author Response
I still think that this is an incremental work with respect to what is reported in the literature on the same topic and the novelty is therefore very limited.
A 3D printing-based manufacturing method using biomass-fungi composites was first reported in 2020 [9]. Effects of mixture composition on rheological properties of the mixture were reported in 2021 [10]. The literature does not have reported results on the effects of waiting time after mixture preparation on mechanical properties, rheological properties, minimum extrusion pressure, and print quality of the prepared mixture. This paper will fill this gap.
"Optical microscopy": it means studying the surface morphology of the 3D-printed sample. This is not very difficult to do and could be of interest to the readers, as one of the main issues related to 3D printing is the quality of the surface of the obtained printed parts. Indeed, the morphology and surface quality of the 3D printed samples is completely disregarded,
The authors agree with the reviewer that (1) it is not very difficult to use optical microscopy to study the surface morphology of printed samples, (2) results from optical microscopy could be of interest to readers, and (3) one of the main issues related to 3D printing is the quality of the surface of the obtained printed parts. The authors will utilize optical microscopy to study surface morphology of printed samples in the future and report the results in a separate paper.

Round 2
Reviewer 1 Report
The paper is suitable for the publication.
Reviewer 2 Report
Dear Authors,
I accept the corrections made.
Reviewer 3 Report
The novelty proposed by the authors is still very limited. However, the manuscript can be published in its present form.
This manuscript is a resubmission of an earlier submission. The following is a list of the peer review reports and author responses from that submission.
Round 1
Reviewer 1 Report
The paper Effects of Waiting time between mixture preparation and printing in 3D printing of Biomass-fungi composites could potentially be of interest to the reader, but the work is too poor and "artisanal" to be published.
The results and the cases studied are very limited. The tools used (for example a ruler to measure the dimensions of the layers Fig. 6) are not adequate. The authors will have to make the work more scientific and in-depth to be published
Reviewer 2 Report
Dear Authors,
Overall, this is an interesting manuscript and I have read it with great interest. However, you need to correct the entire manuscript in a few points. This mainly applies to the discussion of research results.
Detailed comments below:
Introduction: The introduction is too short. I think that you should still show the innovation of mycelium biocomposites against the standard methods of biocomposites production, e.g. by hot pressing. This method also uses by-products from agri-food processing. Browse the articles below: "Properties of biocomposites from rapeseed meal, fruit pomace and microcrystalline cellulose made by press pressing: Mechanical and physicochemical characteristics", or "Sustainable Biocomposites from Poly (butylene succinate) and Apple Pomace: A Study on Compatibil Performanceization".
In general, you can also mention other methods of producing biocomposites. The pouring method is used to produce eg biodegradable TPS films for agricultural applications, eg from agricultural biogas plant waste "Properties of Biocomposites Produced with Thermoplastic Starch and Digestate: Physicochemical and Mechanical Characteristics". Only on this basis can you present the advantages of your method in detail. This will better highlight the innovation of your research.
At the end of the introduction, you should also be precise about the purpose of your work.
Methodology: The description of the materials should include: (name of the raw material: manufacture, city, country)
Table 1: Add the accuracy of the measurements +/- each for the component. Generally this is usually due to the accuracy of the weight used.
Specify what kind of wheat flour you used (enter some parameters, even commercial ones, e.g. name, type 500 ?, etc.)
In the case of apparatus, the description should also be used (model: manufacturer, country, city). This applies to any apparatus used in research.
Results and Discussion: There is basically no discussion of the research results in this chapter. You should try to compare your results to the studies of other scientists (even other materials). Add more citations. You can use e.g. 3-D printing of PLA material. It is not known if your results are near or far from the expected value. Sample article "Effect of print speed and extrusion temperature on properties of 3D printed PLA using fused deposition modeling proces".
Review this chapter and make adjustments at each stage of this chapter.
Conclusion: Basically these conclusions are more of a study statement. Also try to add a more perspective conclusion. For example, it may be influenced in the future by this technology on the development of …… something.
References: You have to adapt the references to the journal requirements and add more references.
Reviewer 3 Report
The paper from Rahman et al. investigates the effects of waiting time between mixture preparation and printing in 3D Printing of biomass-fungi composites. The novelty is quite limited, as the same system has been already studied by the same group in two previous papers (J. Manuf. Mater. Process. 2021, 5(4), 112; Manuf. Lett. 2020, 24, 96–99.), which focused on specific processing parameters. Therefore, it is an incremental work with respect to the previous studies. Besides, the scientific quality of the manuscript is very poor and the conclusions are only marginally supported by the experimental data.
Further comments:
- rheological measurements should be performed on the samples "aged" for different times from the preparation step and the results obtained discussed
- some optical microscopy analyses should be performed and the obtained results commented
- mechanical tests (compression tests) should be performed at different times from the preparation step. The definition of hardness test made by the authors is very confusing and not scientifically sound
Reviewer 4 Report
The manuscript submitted by Rahman et al. presents a method to extrude and manufacture 3D shapes out of biomass-fungi composites. The study shows potential for being relevant to the research field of biocomposites. However, it needs further amendments before it is ready to be accepted. I hereby include my suggestions to improve the quality of the work:
1) The title should be rephrased to increase the scientific soundness of the work.
2) Abstract: the mentioned reduction of material wastage is related to what specifically? For instance, are the authors referring to reducing the internal waste of material post-production or providing sustainable solutions to current processes/materials?
3) Abstract: there was no waiting time of 0 h in this study; please rephrase.
4) Please provide accurate figures to the data presented in the introduction as the motivation of the work. For instance, what is the specific negative impact on the environment of the packaging industry? This should also be coupled with relevant references.
5) In the introduction, a review is given on the recent works on mycelium composites but not on specific applications demonstrated by these materials. I suggest the authors refer to relevant references where the application of such biocomposites has been verified. I recommend the following as a guide: https://doi.org/10.1016/j.carbpol.2021.118477.
6) Extrusion of biomass is a novel field, and this work shows promising results for such a field. However, in the introduction, no relevant references provide insights on the current state-of-the-art of biomass extrusion from an overall perspective. I suggest the authors include such a short reflection. Some literature is here presented: https://doi.org/10.3390/polym12020459, https://doi.org/10.3390/ijms151018967
7) Figure 1 is very comprehensive and self-explanatory. Is it reproduced from reference 11, or has it been re-drawn?
8) Are there any results of the final product? It sounds like the work only provides results until Stage 4 (Figure 1). Please, clarify.
9) What is the “biomass-fungi material”? The wheat flour is regular, for bakery, whole?
10) Are the compositions shown in Table 1 obtained from optimizations or previous work? Please, specify.
11) Is there any pressure gauge at the extruder die and/or measurement of the force of the screw in the extruder used?
12) The 5 experiments of the compression test, do they represent 5 different specimens or 5 measurements on the same specimen? Please, clarify.
13) How do you make sure that the force applied to the cup and plunger for the sample preparation (Figure 3) is always the same?
14) Why does figure 9 not have error bars?
15) Can the authors explain why the hardness is increasing with the increasing waiting time?
16) More discussion on the results is needed to be included in the results and discussions.
17) The conclusions need to be clearer and guide the reader on how this study is novel and applicable for future work.